# A Novel Type of Boundary Extraction Method and Its Statistical Improvement for Unorganized Point Clouds Based on Concurrent Delaunay Triangular Meshes

**DOI:** 10.3390/s23041915

**Published:** 2023-02-08

**Authors:** Xiuzhi He, Rongqi Wang, Chao Feng, Xiaoqin Zhou

**Affiliations:** 1Key Laboratory of CNC Equipment Reliability, Ministry of Education, Changchun 130025, China; 2School of Mechanical and Aerospace Engineering, Jilin University, Changchun 130025, China

**Keywords:** point cloud data, 3D laser scanning, boundary extraction, Delaunay triangular meshes, statistical characteristics

## Abstract

Currently, three-dimensional (3D) laser-scanned point clouds have been broadly applied in many important fields, such as non-contact measurements and reverse engineering. However, it is a huge challenge to efficiently and precisely extract the boundary features of unorganized point cloud data with strong randomness and distinct uncertainty. Therefore, a novel type of boundary extraction method will be developed based on concurrent Delaunay triangular meshes (CDTMs), which adds the vertex-angles of all CDTMs around a common data point together as an evaluation index to judge whether this targeted point will appear at boundary regions. Based on the statistical analyses on the CDTM numbers of every data point, another new type of CDTM-based boundary extraction method will be further improved by filtering out most of potential non-edge points in advance. Then these two CDTM-based methods and popular *α*-shape method will be employed in conducting boundary extractions on several point cloud datasets for comparatively analyzing and discussing their extraction accuracies and time consumptions in detail. Finally, all obtained results can strongly demonstrate that both these two CDTM-based methods present superior accuracies and strong robustness in extracting the boundary features of various unorganized point clouds, but the statistically improved version can greatly reduce time consumption.

## 1. Introduction

In the past two decades, three-dimensional (3D) laser scanning technology and its equipments have rapid developments and broad applications in many important fields, such as the reverse engineering [1], non-contact measurements [2], computer vision [3], and building reconstruction [4]. Point cloud dataset collected by 3D laser scanners is one of the most basic and popular data types in engineering practices [5]. However, such 3D laser-scanned point cloud datasets are commonly unorganized, with strong randomness and distinct uncertainty, which brings a huge difficulty to rapidly and precisely extract the boundary features of various complex-shaped point cloud data. Therefore, it is very essential to develop fast and exact boundary extraction methods, since they will directly determine the accuracies and efficiencies of various industrial applications.

Currently, boundary extraction methods of point clouds could be mainly classified into two categories: the methods based on geometric features of scattered point clouds, and the methods based on triangular meshes of point clouds. Among the first kind of boundary extraction approaches, the K-nearest neighbor (K-NN) method and its variants have been one of the most widely used algorithms in extracting the boundary features of point cloud data [6,7,8,9,10]. The core principle of K-NN method is to firstly search K nearest neighbor points around a certain targeted data point, then to mathematically characterize the spatial distribution of these searched points to judge whether this targeted data point is located at boundary region or not. In general, the K-NN boundary extraction method must define extra estimating indicators to describe the distribution features of K searched points, such as the standard deviations of Euler distances or the included angles among the targeted point and K searched points. Unfortunately, the threshold selections of these estimating indicators depend highly on the local distribution characteristics of scattered point clouds, such as their uniformities and densities. Thus, it is totally non-intuitive and very difficult to select an optimal threshold that can guarantee a fine extraction accuracy for unorganized point clouds with highly uneven and strongly random distribution [11]. Although multiple different estimating indicators can be commonly adopted to conduct the boundary extractions on various complex-shaped point clouds for higher accuracies, their threshold selections are still very complex and challenging in practices. 

As an improvement, the alpha shape (*α*-shape) algorithm proposed by Edelsbrunner et al. [12] is another popular alternative. It can abstract the intuitive shape of the point clouds by traversing the dataset of discrete points with a circle of fixed radius *α*, guided by above principle that only every pair of two points lined on the circle of fixed radius without any other extra data points can be judged as the boundary points. This algorithm has been more broadly used to extract inner and outer boundary features of convex and concave polygons [13]. However, similar to K-NN, *α*-shape method also faces the optimal threshold selection issue. In other words, the extraction performance of *α*-shape method highly depends on its key tunable parameter *α*. If α is too large, the boundary features and internal details are easily lost; on the contrary, if α is too small, more discrete patches and holes will appear adversely. 

The second kind of methods of boundary extraction are based on triangular meshes. The main advantage of triangular meshes for boundary extraction is that it can process the scattered and disordered point clouds into topological structure form to enhance its semantic information and extract boundary more accurately. Many existing studies have been explored for boundary extraction based on triangular meshes [14,15,16]. Among them, the most commonly used mesh of triangulation is Delaunay triangular meshes (DTMs), in which the circumcircle of a triangle contains no point in its interior [17]. Nevertheless, when employing the DTMs to simply and rapidly extract the boundary features of point cloud data, some imperfect DTMs with abnormal sizes and irregular shapes are ignored unfavorably, which inevitably cause a certain level of deteriorations of extraction validity and precision due to these imperfect DTMs may be located at boundary regions [16]. 

In summary, various existing boundary extraction methods generally exhibit three aspects of distinct deficiencies. Firstly, the extraction accuracies of K-NN method and its variants are highly sensitive to the spatial or planar distribution features of point clouds, their obvious uncertainty and randomness are very easily to cause the error-judgments of non-edge points (False positive) and the miss-judgments of edge points (False negative) in certain regions. The second one is the unacceptable extraction efficiencies. For example, both K-NN, *α*-shape and their all variants must separately conduct the tedious extracting operations on each data point across whole dataset. This will definitely cause excessive time consumption and poor extraction efficiency when we need to process huge amounts of point cloud dataset. Finally, K-NN, *α*-shape and their variants must select appropriate thresholds (such as K-value and *α*-value) to estimate whether target data points appear at boundary regions or not. However, the threshold selections are totally non-intuitive and troublesome due to the terrible uniformity and distinct uncertainty of unorganized point cloud datasets. Therefore, it is very difficult to select optimal estimating thresholds that can guarantee high precision and strong robustness of boundary extractions in practices, especially for those unorganized point cloud dataset that must suffer strong randomness, distinct uncertainty and terrible uniformity. 

In view of above-mentioned facts, this study will develop a novel type of boundary extraction algorithm for unorganized point cloud data based on the Delaunay triangular meshes that share a data point as their common vertex, namely the concurrent Delaunay triangular meshes (CDTMs), then the degree sum of all CDTM vertex-angles around this common-shared data point will be innovatively defined as a critical evaluating indicator to judge whether target data point is located at boundary region or not. Considering that most edge points generally have fewer CDTMs than non-edge points, the total number of CDTMs around each data point can be statistically analyzed to further improve primitive CDTM-based boundary extraction method for higher efficiency, termed as the statistical CDTM-based boundary extraction method. Finally, a series of numerical simulations and practical experiments will be conducted on different complex-shaped point cloud dataset to distinctly reveal the effectiveness, feasibility and robustness of these two CDTM-based methods, as well as performing a fair comparison with popular *α*-shape method.

In short, the main contributions of this study can be briefly summarized as below: 

(a) A novel CDTM-based boundary extraction method will be developed for exactly characterizing the geometric boundaries of unorganized point cloud dataset with strong randomness and terrible uniformity, and which does not have to face the actual dilemma of optimal threshold selections similar to K-NN and *α*-shape method.

(b) Based on the statistical analyses on CTDM numbers of edge points and non-edge points, another CDTM-based boundary extraction method will be further improved for near extraction accuracy and less computational burden through pre-filtering out those potential non-edge points that possess more than five CDTMs.

(c) Based on simulation and experimental scenarios, fair comparisons with popular α-shape method will be conducted on different point clouds of several complex-shaped workpieces and a mechanical gear, all obtained results can well demonstrate the superior performances of the two newly proposed CDTM-based methods.

The remainder of this paper can be briefly organized as follows: Section 2 will firstly describe the basic principles of new CDTM-based boundary extraction method in detail, its crucial performances will be also investigated and compared with popular *α*-shape method. Section 3 will develop another improved version with much higher extraction efficiency, namely statistical CDTM-based boundary extraction method, also the similar performance comparison will be conducted on same point clouds. In Section 4, the point cloud datasets of complex-shaped workpieces will be experimentally collected through a 3D laser scanner to further investigate and compare the extraction performances of these two CDTM-based methods and *α*-shape method. Finally, several dominant conclusions and further research directions will be summarized and discussed in Section 5. 

## 2. Fundamental Principles of CDTM-Based Boundary Extraction Method

According to the generation principle of Delaunay triangular meshes (DTMs) and its improved versions, the fundamental of CDTM-based boundary extraction method can be mathematically constructed with the Euler distance and Cosine theorem. Afterwards, the unorganized point cloud dataset of two different complex-shaped objects are employed for verifying its boundary extraction performances involving efficiency and accuracy.

### 2.1. Modification of Delaunay Triangulation

In general, the triangular mesh generation of complex-shaped geometries has been one of the most critical procedures in lots of industrial applications [17,18,19], especially for the boundary extraction of unorganized point clouds. As one of the most extensive mesh triangulation method, the Delaunay triangulation algorithm and its improved versions has been widely adopted to solve various complex problems in the past few decades [19]. In view of the generated Delaunay triangular meshes will be completely non-overlapped, and there will be no internal data points can be found in the circumcircle of an arbitrary triangular mesh. As a result, Delaunay triangulation can maximize the minimum interior angles among all potential triangular meshes and avoid the undesired slender triangles as much as possible [19,20]. 

The Delaunay triangular meshes (DTMs) are generated on a point cloud dataset of complex object, as shown in Figure 1. It can be obviously found that these DTMs can be only applicable to the convex closure geometries that consist of no concave substructures, such as holes and cavities. Unfortunately, the great majority of practical objects or parts have inevitably suffered various concave substructures, thus the Delaunay triangulation method cannot be directly applied to the boundary extractions in most practical cases. To be more specific, this is mainly because that the concave substructures may be fully filled by a series of imperfect and undesired DTMs with abnormal sizes and irregular shapes, as shown in Figure 1. This will ultimately cause the vital information loss of boundary characteristics and the sharp deterioration of extraction accuracies. 

Therefore, it is very necessary to correctly remove the undesired Delaunay triangle meshes that have irregular shapes and abnormal sizes before using boundary extraction method. Inspired by existing evaluation methods for triangular mesh qualities, which are broadly applied in finite element analysis (FEA), the aspect ratio (*AR*) is firstly suggested as a major evaluating indicator to filter out most undesired DTMs, mainly involving the slender triangular meshes. Here, we suppose the three side-lengths of the *k*-th Delaunay triangular mesh are $d_{1}^{k}$, $d_{2}^{k}$ and $d_{3}^{k}$, respectively, its corresponding aspect ratio *AR^k^* can be expressed as follows [17]: (1)$$AR^{k}=\frac{d_{1}^{k}\cdot d_{2}^{k}\cdot d_{3}^{k}}{8\left(s^{k}-d_{1}^{k}\right)\left(s^{k}-d_{2}^{k}\right)\left(s^{k}-d_{3}^{k}\right)},s^{k}=\frac{d_{1}^{k}+d_{2}^{k}+d_{3}^{k}}{2}.$$

The *AR* of a valid triangular mesh should less than two thresholds suggested by [20], but the two thresholds defined in Equation (1) must be properly regulated with certain practical conditions. Meanwhile, the minimum value $\theta_{\min}^{k}$ among three interior angles of the *k*-th triangle will be defined to remove the slender triangular meshes that frequently appear at boundary regions of point cloud dataset. After conducting above removal processes with appropriate threshold selection of $\theta_{\min}^{k}$, the modified DTMs will contain very few slender triangular meshes, but a certain amount of oversize triangular meshes can be observed at inner holes and cavity regions of point cloud dataset, whose average side-lengths are much longer than those of valid triangular meshes, which need to be further removed. Therefore, the average value of three side-lengths of the *k*-th triangle can be also assumed as another key evaluation indicator (abbreviated as $d_{ave}^{k}$) to filter out the oversize triangular meshes. According to the distribution density of point cloud dataset, the threshold of $d_{ave}^{k}$ can be actually selected as three or more times of average side-lengths of all triangle meshes. Note that the threshold selections of *AR^k^*, $\theta_{\min}^{k}$ and $d_{ave}^{k}$ should be strictly regulated with distribution features of point clouds, this is due to the intrinsic differences and random features of unorganized point clouds, the modified DTMs after removing adverse triangular meshes are finally illustrated in Figure 2.

Comparing with the initially generated DTMs shown in Figure 1, it can be clearly seen that the modified DTMs can directly emerge the boundary features of point cloud dataset of sampled complex-shaped object with concave substructures, such as holes and cavities. Therefore, how to rapidly and exactly obtain the boundary features from above modified DTMs will become the most pivotal procedure of the CDTM-based boundary extraction method developed in this study, which will be introduced in the following subsections.

### 2.2. Mathematical Derivations of CDTM-Based Boundary Extraction Algorithm

The schematic diagram of this proposed CDTM-based boundary extraction method is illustrated in Figure 3, the basic principles are described as below: Firstly, each data point will be, respectively, regarded as a common-vertex that is concurrently shared by several DTMs. In other words, these so-called concurrent Delaunay triangular meshes (CDTMs) can adopt a same data point as one of their three vertexes. Afterwards, the total degree of all vertex-angles around this adopted data point will be added together as an estimation indicator to judge whether this data point appears at boundary region or not. If the total degree of all vertex-angles is perfectly equal to 360°/2π, namely a perigon or round angle, this data point will be identified as a non-edge point; otherwise, it will be identified as an edge point. Obviously, this defined estimation indicator will contain two advancements: Firstly, the degree sum of vertex-angles will very intuitively employ a constant threshold (360° or 2π), thus no threshold optimization needs to be additionally conducted such as K-NN, α-shape method and their improved versions. Secondly, the threshold selections and extraction accuracies will be completely independent on the distribution features of unorganized point cloud data, such as uniformity and density, which will be conducive to greatly strengthen the robustness of boundary extractions.

Next, the basic principles of proposed CDTM-based boundary extraction method and its mathematical derivations will be further described as follows.

#### 2.2.1. Mathematical Basics of Concurrent Delaunay Triangular Meshes (CDTMs)

As illustrated in Figure 3, the generated CDTMs for unorganized point cloud dataset collected by a 3D laser scanner can be mathematically described with a triangular mesh matrix ***M***(*K*, 3), which consists of all data points and corresponding spatial position coordinates, but here only considers as 2D point cloud data, as formulated in following:(2)$$\begin{array}{cc} M\left(K,3\right)=\left[\begin{array}{c} M^{1}\\ \vdots \\ M^{k}\\ \vdots \\ M^{K} \end{array}\right]=\left[\begin{array}{ccc} p_{1}^{1}\left(x,y\right) & p_{2}^{1}\left(x,y\right) & p_{3}^{1}\left(x,y\right)\\ \vdots & \vdots & \vdots \\ p_{1}^{k}\left(x,y\right) & p_{2}^{k}\left(x,y\right) & p_{3}^{k}\left(x,y\right)\\ \vdots & \vdots & \vdots \\ p_{1}^{K}\left(x,y\right) & p_{2}^{K}\left(x,y\right) & p_{3}^{K}\left(x,y\right) \end{array}\right]; & k=1,\;2,\;\ldots ,\;K. \end{array}$$
where, $M^{k}=\left[\begin{array}{ccc} p_{1}^{k}\left(x,y\right) & p_{2}^{k}\left(x,y\right) & p_{3}^{k}\left(x,y\right) \end{array}\right]$ denotes for the *k*-th Delaunay triangular mesh, the planar position coordinates of its three vertices can be expressed as $p_{1}^{k}\left(x,y\right)$, $p_{2}^{k}\left(x,y\right)$ and $p_{3}^{k}\left(x,y\right)$, respectively. Obviously $p_{1}^{k}\left(x,y\right)\neq p_{2}^{k}\left(x,y\right)\neq p_{3}^{k}\left(x,y\right)$. *K* stands for the total number of the modified Delaunay triangular meshes on point cloud dataset.

Next, the three side-length $D^{k}=\left[\begin{array}{ccc} d_{1}^{k} & d_{2}^{k} & d_{3}^{k} \end{array}\right]$ of the *k*-th triangle mesh *M^k^* can be, respectively, calculated based on the Euler distance formulations, as expressed as below:(3)$$\begin{array}{cc} D^{k}={\left[\begin{array}{c} d_{\begin{array}{l} 1 \end{array}}^{k}\\ d_{\begin{array}{l} 2 \end{array}}^{k}\\ d_{3}^{k} \end{array}\right]}^{T}={\left[\begin{array}{c} {\left\|p_{1}^{k}-p_{2}^{k}\right\|}_{\begin{array}{l} 2 \end{array}}\\ {\left\|p_{2}^{k}-p_{3}^{k}\right\|}_{\begin{array}{l} 2 \end{array}}\\ {\left\|p_{3}^{k}-p_{1}^{k}\right\|}_{2} \end{array}\right]}^{T}={\left[\begin{array}{c} \sqrt{{\left(x_{1}^{k}-x_{2}^{k}\right)}^{2}+{\left(y_{1}^{k}-y_{2}^{k}\right)}^{2}}\\ \sqrt{{\left(x_{2}^{k}-x_{3}^{k}\right)}^{2}+{\left(y_{2}^{k}-y_{3}^{k}\right)}^{2}}\\ \sqrt{{\left(x_{3}^{k}-x_{1}^{k}\right)}^{2}+{\left(y_{3}^{k}-y_{1}^{k}\right)}^{2}} \end{array}\right]}^{T}; & k=1,\;2,\;\ldots ,\;K. \end{array}$$
where $\left(x_{1}^{k},y_{1}^{k}\right)$, $\left(x_{2}^{k},y_{2}^{k}\right)$ and $\left(x_{3}^{k},y_{3}^{k}\right)$ denote the planar position coordinates of three vertexes of the *k*-th Delaunay triangular meshes, respectively. Then the three side-length matrix ***D***(*K*, 3) of all triangular meshes can be further calculated as follows:(4)$$\begin{array}{cc} D\left(K,3\right)=\left[\begin{array}{c} D^{1}\\ \vdots \\ D^{k}\\ \vdots \\ D^{K} \end{array}\right]=\left[\begin{array}{ccc} d_{1}^{1} & d_{2}^{1} & d_{3}^{1}\\ \vdots & \vdots & \vdots \\ d_{1}^{k} & d_{2}^{k} & d_{3}^{k}\\ \vdots & \vdots & \vdots \\ d_{1}^{K} & d_{2}^{K} & d_{3}^{K} \end{array}\right]; & k=1,\;2,\;\ldots ,\;K. \end{array}$$

Based on above side-length matrix ***D***(*K*, 3), their corresponding vertex-angle matrix ***Φ***(*K*, 3) for all Delaunay triangular meshes ***M***(*K*, 3) can be also calculated as below:(5)$$\begin{array}{cc} \Phi \left(K,3\right)=\left[\begin{array}{c} \Phi^{1}\\ \vdots \\ \Phi^{k}\\ \vdots \\ \Phi^{K} \end{array}\right]=\left[\begin{array}{ccc} \varphi_{1}^{1} & \varphi_{2}^{1} & \varphi_{3}^{1}\\ \vdots & \vdots & \vdots \\ \varphi_{1}^{k} & \varphi_{2}^{k} & \varphi_{3}^{k}\\ \vdots & \vdots & \vdots \\ \varphi_{1}^{K} & \varphi_{2}^{K} & \varphi_{3}^{K} \end{array}\right]; & k=1,\;2,\;\ldots ,\;K. \end{array}$$
where the vertex-angle matrix $\Phi^{k}=\left[\begin{array}{ccc} \varphi_{1}^{k} & \varphi_{2}^{k} & \varphi_{3}^{k} \end{array}\right]$ denotes the three vertex angles of the *k*-th Delaunay triangular mesh *M^k^*, which can be mathematically expressed based on the Cosine theorem, which is described as follows:(6)$$\begin{array}{cc} \Phi^{k}={\left[\begin{array}{c} \varphi_{\begin{array}{l} 1 \end{array}}^{\begin{array}{l} k \end{array}}\\ \varphi_{\begin{array}{l} 2 \end{array}}^{k}\\ \varphi_{\begin{array}{l} 3 \end{array}}^{k} \end{array}\right]}^{T}={\left[\begin{array}{c} \arccos\left(\frac{{\left(d_{1}^{k}\right)}^{2}+{\left(d_{3}^{k}\right)}^{2}-{\left(d_{2}^{k}\right)}^{2}}{2\cdot d_{1}^{k}\cdot d_{3}^{k}}\right)\\ \arccos\left(\frac{{\left(d_{1}^{k}\right)}^{2}+{\left(d_{2}^{k}\right)}^{2}-{\left(d_{3}^{k}\right)}^{2}}{2\cdot d_{1}^{k}\cdot d_{2}^{k}}\right)\\ \arccos\left(\frac{{\left(d_{2}^{k}\right)}^{2}+{\left(d_{3}^{k}\right)}^{2}-{\left(d_{1}^{k}\right)}^{2}}{2\cdot d_{2}^{k}\cdot d_{3}^{k}}\right) \end{array}\right]}^{T}; & k=1,\;2,\;\ldots ,\;K. \end{array}$$

Based on Equations (1)~(6), the key side-length matrix ***D***(*K*, 3) and its corresponding vertex-angle matrix ***Φ***(*K*, 3) for all modified CDTMs can be easily obtained, respectively, which are the important necessaries to develop a novel type of CDTM-based boundary extraction method and its statistically improved version for unorganized point clouds with complex shapes and random distributions.

#### 2.2.2. Basic Procedures of CDTM-Based Boundary Extraction Method

Figure 4 is the basic procedures of the proposed CDTM-based boundary extraction method and its variant, whose specific details will be described through the following steps:

**Step 1:** Collect the raw 3D point data through a laser scanner and convert them into simpler 2D point data through a series of pre-processing operations (such as filtrations and repairs) as well as coordinate transformations (such as translations, rotations and projections).

**Step 2:** Generate the Delaunay triangular meshes (DTMs) on above pre-processed 2D point cloud data, then to removal those undesirable DTMs that possess abnormal size and irregular shapes based on the screening conditions mentioned in Section 2.1, finally to generate modified DTMs and calculate their matrix ***M***(*K*, 3) described by Equation (2).

**Step 3:** Search all DTMs that commonly employ a data point P*_n_* (*n* = 1, 2, …, *N*_0_) as one of their three vertexes, which can be mathematically described as follow:(7)$$\begin{array}{cc} \left[I_{n},J_{n}\right]=\left[\begin{array}{cc} i_{n}^{1} & j_{n}^{1}\\ \vdots & \vdots \\ i_{n}^{r} & j_{n}^{r}\\ \vdots & \vdots \\ i_{n}^{R_{n}} & j_{n}^{R_{n}} \end{array}\right]=\mathrm{find}\left(M\left(K,3\right)=P_{n}\right)=\mathrm{find}\left(\left[\begin{array}{ccc} p_{1}^{1} & p_{2}^{1} & p_{3}^{1}\\ \vdots & \vdots & \vdots \\ p_{1}^{k} & p_{2}^{k} & p_{3}^{k}\\ \vdots & \vdots & \vdots \\ p_{1}^{K} & p_{2}^{K} & p_{3}^{K} \end{array}\right]=P_{n}\right); & \left\{ \begin{array}{l} I_{n}\in \left[1,\;2,\;\ldots ,\;K\right]\\ J_{n}\in \left[1,\;2,\;3\right]\\ n=1,\;2,\;\ldots ,\;N_{0}. \end{array}\right. \end{array}$$
where, **I***_n_* and **J***_n_* denote the serial number set [1, 2, …, *K*] of CDTMs around data point P*_n_* and the serial number set [1, 2, 3] of corresponding mesh vertices where data point P*_n_* just appear at, respectively, namely the row and column position of data point P*_n_* in modified DTM matrix ***M***(*K*, 3). For example, [**I***_n_*, **J***_n_*] = [142, 2; 367, 3; 548, 1]^T^ represents that the *n*-th data point P*_n_* will can be concurrently found at the 2nd vertex of the 142th CDTM, the 3rd vertex of the 367th CDTM and the 1st vertex of the 548th CDTM. Obviously, *R_n_* denotes the total number of all CDTMs that commonly share the *n*-th data point P*_n_* as one of their three vertexes. In addition, the mathematical description of self-defined function *find*(•) is to rapidly search the eligible elements in input matrix and output corresponding row and column position.

**Step 4:** Calculate the vertex-angle sum Θ*_n_* of CDTMs around the *n*-th data point P*_n_*. Θ*_n_* is defined as the total degree of all CDTM’s vertex-angles **Φ**(**I***_n_*, **J***_n_*) that are located at the *n*-th data point P*_n_* simultaneously, as shown in Figure 3, which will be mathematically described as below:(8)$$\begin{array}{cc} \Theta_{n}=\sum_{r=1}^{R_{n}}\theta_{n}^{r}=\sum \Phi \left(I_{n},J_{n}\right)=\sum_{r=1}^{R_{n}}\Phi \left(i_{n}^{r},j_{n}^{r}\right), & n=1,2,\cdots ,N_{0} \end{array}$$
where $\theta_{n}^{r}$ (*r* = 1, 2, …, *R_n_*) stands for the vertex angle of the *r*-th CDTM, which is located at the *n*-th data point P*_n_*.

**Step 5:** Judge the target data point P*_n_* is an edge point or non-edge point and output results. The vertex-angle sum Θ*_n_* introduced in Step 4 will be regarded as a very intuitive evaluation indicator to rapidly and correctly judge whether the data point P*_n_* appear at boundary regions of unorganized point clouds or not, as described by Equation (9).
(9)$$\left\{ \begin{array}{l} \Theta_{n}=360^{\circ };\;P_{n}\;\mathrm{is}\;a\;\mathrm{non-edge}\;\mathrm{point}\\ \Theta_{n}<360^{\circ };\;P_{n}\;\mathrm{is}\;a\;\mathrm{edge}\;\mathrm{point} \end{array}\right.$$

In theory, there will be no error-judgment of non-edge point (False positive) and few miss-judgments of edge points (False negative) can be clearly observed through adopting above evaluation index Θ*_n_*, this is due to its interdependences on the uncertain distribution and non-uniform density of unorganized point cloud data. More importantly, this critical index Θ*_n_* has no extra parameter or threshold that must be tediously modulated based on the distribution features of point clouds, such as the K-values and α-values in K-NN and α-shape methods, thus which can make the CDTM-based boundary extraction processes are totally intuitive, very user-friendly and strongly robust for different point cloud data with uncertain distribution and non-uniform density.

### 2.3. Simulation and Comparison of Proposed CDTM-Based Method

In order to validate the feasibility and effectiveness of above proposed CDTM-based boundary extraction algorithm, this section will conduct a series of boundary extractions on the point cloud dataset of two different typical objects, namely the square-circular (SC) shape and circular-square (CS) shape objects, respectively. The commercial mathematics software MATLAB 2015b (version: 8.6.0.267246) and its Delaunay toolbox are adopted to program the CDTM-based boundary extraction method through using a computer with operating system Windows 7, which is configured with Intel Core i5-4590 CPU@3.30 GHz and 16 GB RAM. Finally, the obtained extracting results are shown in Figure 5.

For SC-shaped object with 8, 853 data points, this developed CDTM-based method can rapidly and accurately extract 557 edge points with 1.2 s time consumption, as shown in Figure 5a,b. Similarly, 514 edge points can be effectively extracted from the 5, 873 data points of CS-shaped object with 0.76 s time consumption, as illustrated in Figure 5c,d. Meanwhile, there are no error-judgment of non-edge points (False positive) and very few miss-judgments of edge points (False negative) can be observed both in above extraction results on SC-shaped and CS-shaped objects, which can well demonstrate that this novel CDTM-based boundary extraction method can rapidly and exactly search out almost all valid edge points in unorganized point clouds. Unlike popular K-NN, *α*-shape methods and their various variants, no parameter and threshold need to be additionally calibrated and regulated for optimizing their extraction efficiencies and accuracies in CDTM-based boundary extraction processes.

Similarly, the popular *α-*shape method with different *α-*values are also employed for extracting the boundary features of above two point cloud datasets, respectively, as well as comparing with the proposed CDTM-based method, as shown in Figure 6.

For SC-shaped object, 560 edge points can be found out when *α-*value is set as *α* = 0.5, as shown in Figure 6a, which is slightly more than the extraction results (557 edge points) shown in Figure 5a, this is because that three non-edge points are misjudged as three edge points in a local non-boundary region, namely three false positive points. As shown in Figure 6b, 557 edge points can be perfectly extracted when *α-*value is chosen as *α* = 0.6, which is completely equal to the total number of edge points extracted by CDTM-based method. Comparing with the extraction results shown in Figure 5b, two miss-judgments of edge points (False negative) will be gradually found at certain boundary region when *α-*value increases up to 0.7, as shown in Figure 6c. Similarly, the *α-*shape method is also adopted to extract the boundary features of SC-shaped point cloud data, it turns out that the *α-*value has similar influence rules on the total numbers of false positive points and false negative points, as illustrated in Figure 6d–f. Concretely, the total number of false positive points will gradually decrease with increasing *α-*values, but the total number of false negative points will increase at the same time. Relative to CDTM-based method, *α-*shape method must search an optimum threshold to guarantee the highest extraction accuracy according to the distribution uniformity and density of point cloud datasets, but it is totally non-intuitive and very difficult owing to the influences of strong randomness and distinct uncertainty. It is very important to note that the *α-*shape method with three different *α-*values may expend much less time consumptions (about 0.2 s) than proposed CDTM-based method, this is mainly due to the *α-*shape method can directly call some built-in functions (such as alphaShape and boundaryFacets functions) in MATLAB 2015b. However, the CDTM-based boundary extraction method should have approximate time consumptions with *α-*shape method because of they have similar time complexities.

## 3. Statistical Improvement of CDTM-Based Boundary Extraction Method

In previous section, the new CDTM-based boundary extraction method was proven to accurately and effectively describe the boundary features of unorganized point clouds. However, similar to other existing boundary extraction methods such as the K-NN and *α*-shape methods, this CDTM-based method must continuously conduct the extracting operations on every data point in whole cloud datasets. Obviously, this inevitably causes the sharp increases in computational burden and time consumption, greatly restricting practical applications in various important fields which demand extremely high real-time properties. Therefore, it is necessary to improve above CDTM-based boundary extraction method for higher efficiency and faster calculation.

From detailed analyses in Section 2.2 and triangulation results of complex-shaped object shown in Figure 2, we can easily observe that each data point shares a different number of CDTMs, however, the CDTM numbers of edge points are generally fewer than non-edge points. More specifically, it can be statistically found that the majority of edge points possess four CDTMs or less, on the contrary, most of the non-edge points usually share five or more CDTMs. Inspired by these important observations, the CDTM number of each data point can be employed as a statistical evaluation indicator to strictly filter out the majority of potential non-edge points with five or more CDTMs in advance, further to improve the extraction efficiency of new CDTM-based method. For more quantitative discussions, the total numbers of data points with different numbers of CDTMs will be statistically analyzed and compared on the point cloud datasets adopted in Section 2.3. Finally, the obtained statistical results are illustrated in Figure 7a,b, respectively.

In summary, both more than 57% of SC-shaped and CS-shaped data points share six CDTMs (namely *δ_N_* = 6), meanwhile, more than 90% of data points possess five CDTMs or more (namely *δ_N_* ≥ 5), as illustrated in Figure 7. For SC-shaped object, the total number of the data points with no more than three CDTMs (*δ_N_* ≤ 3) is 546, which is very close to the total number of all edge points (557 points) shown in Figure 5a. For CS-shaped object, the total number of the data points with no more than three CDTMs (*δ_N_* ≤ 3) is 507, similarly, which is also slightly less than the total number of all edge points (514 points) shown in Figure 5c. From above statistical analyses, we can distinctly know that almost all of edge points will share no more than three CDTMs, which can well demonstrate the intuitive observations on the dependencies between triangular meshes and data points, as shown in Figure 2. Based on above crucial conclusions, another type of statistical CDTM-based boundary extraction method with near precision and higher efficiency can be developed through preliminarily filtering out massive irrelevant data points (the potential non-edge points) that possess more than a certain number *δ_N_* of CDTMs. As a result, a series of boundary extractions with different *δ_N_* must be firstly conducted to select optimum *δ_N_*, respectively, as shown in Figure 8. 

When CDTM number is selected as *δ_N_* ≤ 3, 546 filtered data points are entirely located at boundary regions, just a very few numbers of edge points are missed, as shown in Figure 8a, corresponding time consumption is only 0.5 s. When CDTM number is set as *δ_N_* ≤ 4, *δ_N_* ≤ 5 and *δ_N_* ≤ 6, all edge points (*N*_1_ = 557) of SC-shaped object can be very perfectly extracted with no false positive (FP) points, as shown in Figure 8b–d, corresponding time consumptions gradually increases from 0.5 s to 1.1 s with increasing *δ_N_* values. Similarly, 507 edge points of CS-shaped object can be also extracted with 0.44 s time consumption when CDTM number is selected as *δ_N_* ≤ 3, which is slightly less than the total number of all edge points (514 points) shown in Figure 5c, this is mainly due to the edge points that have more than three CDTMs (*δ_N_* > 3) are very few, only seven false negative (FN) points can be observed, as shown in Figure 9a. When CDTM numbers are selected as δ*_N_* ≤ 4, *δ_N_* ≤ 5 and *δ_N_* ≤ 6, 514 edge points of CS-shaped object can be extracted with no false positive (FP) points, as illustrated in Figure 9b–d, respectively, their corresponding time consumptions are 0.46 s, 0.54 s and 0.72 s, these results show that the extraction speeds of improved CDTM-based method are faster and acceptable relative to its original version.

For quantitatively analyzing and optimizing the CDTM number *δ_N_*, the total number of edge points and the time consumptions under different *δ_N_* values (from 3 to 6) should be further compared with original CDTM-based method (*δ_N_* = All), as listed in Table 1, the edge extraction accuracy *ε* is defined to be the percentage of actually extracted edge point number *N*_1_ to all edge point number *N*_0_, as expressed by Equation (10). It can be clearly observed that the selections of *δ_N_* value will directly determine the extraction accuracies *ε* and time consumption *T_c_* of statistical CDTM-based method, but the trade-off between extraction accuracy and computational burden need to be further balanced in different practical applications. For the point cloud datasets of SC-shaped and CS-shaped objects, all edge points can be completely extracted when CDTM number is set as *δ_N_* ≤ 4, namely extraction accuracy *ε* = 100%. However, their corresponding time consumptions are 0.5 s and 0.46 s, respectively, which are only 41.6% and 60.5% of original CDTM-based method shown in Figure 5 (1.2 s and 0.76 s). In a short, all obtained results can explicitly indicate that this statistically improved CDTM-based method has near extraction accuracy but less time consumption than its original version, which further convincingly demonstrate that the statistically improved CDTM-based method can present higher effectiveness and stronger feasibility than its original version.
(10)$$\varepsilon =\frac{N_{1}}{N_{0}}\times 100\%$$

## 4. Implementation and Discussion

Firstly, both the original and improved CDTM-based boundary extraction methods are conducted on a set of point cloud data of an involute cylinder gear for measuring its main geometric parameters such as diameter and module, which totally contains 49, 549 data points, as shown in Figure 10a. Based on the modification of Delaunay triangular meshes presented in Section 2.1, the desirable DTMs of mechanical gear can be rapidly generated, the dada points and corresponding DTMs in a 1/4 area A are locally shown in Figure 10b. Similar to the statistical analyses on point cloud datasets of SC-shaped and CS-shaped objects shown in Figure 7, the point cloud dataset of involute cylinder gear is also statistically analyzed in detail, as shown in Figure 10c.

From the obtained results, we can also summarize some crucial conclusions that are similar with the descriptions on Figure 7. For example, the total number of data points that have six CDTMs is up to 42,018, which is about 84.8% of all data points, others will not be repeated here. Afterwards, the main performances such as extraction accuracy and time consumption of above proposed CDTM-based boundary extraction method and its improved version are revealed and discussed in detail, also fairly comparing with popular *α*-shape method. Finally, the obtained results are illustrated in Figure 11. 

When CDTM number is selected as *δ_N_* ≤ 3, only 2342 edge points can be found out with 1.9 s time consumption, about 300 miss-judging edge points (False negative) can be unfortunately observed, as shown in Figure 11a. However, 2622 and 2624 edge points can be extracted when CDTM number is selected as *δ_N_* ≤ 4, *δ_N_* ≤ 5 and *δ_N_* ≤ 6, respectively, corresponding time consumption are 2.1 s, 3.1 s and 21.0 s, as illustrated in Figure 11b–d. Obviously, the total number of extracted edge points is very close to the total number of data points (2803 points) that possess four CDTMs or less (*δ_N_* ≤ 4), as shown in Figure 10c, which is similar with the statistical analyses on point cloud datasets of mechanical gear, SC-shaped and CS-shaped parts. Comparing with original CDTM-based method shown in Figure 11e and α-shape method shown in Figure 11f, the improved version can exhibit an approximate extraction accuracy when CDTM number is set as *δ_N_* ≤ 4, but whose time consumption (2.1 s) only is less 9.6% of its original version (21.8 s). It is very important to note that α-shape method has less time consumption (1.0 s) than statistical CDTM-based method, this is mainly due to α-shape method can directly call the built-in alphaShape and boundaryFacets functions in MATLAB 2015b, they should have approximate time consumption in consideration of their similar time complexities. In addition, the α-shape method (α = 0.8) can find out slightly more edge points than CDTM-based method and its improved version (*δ_N_* ≤ 4), but a few of miss-judging edge points are obviously found in the keyway corners of gear, namely the region B illustrated in Figure 11f, this obtained results can strongly demonstrate that CDTM-based method will have higher extracting precision than α-shape method, this is due to α-shape method with a constant α-value is very easy to cause a certain number of false negative (FN) points and false positive (FP) points.

For more distinctly comparing above three extraction method, their edge extraction accuracy *ε* and other main performances with different *δ_N_* and *α*-values are, respectively, calculated with Equation (10), all obtained results are listed in Table 2.

Next, another set of actual point cloud data of a complex-shaped workpiece will be experimentally collected and adopted to further verify the effectiveness and feasibility of CDTM-based boundary extraction method and its improved version. As shown in Figure 12, a 3D binocular laser scanner with structured light (VTOP200B, Visentech Co., Ltd., Tianjin, China) will be employed for obtaining the point cloud data of a complex-shaped workpiece, whose CCD resolution and scanning precision are 5 mega-pixel and 0.01~0.03 mm, the minimum and maximum formats of single scanning operation are 50 × 38 × 38 mm^3^ and 450 × 342 × 342 mm^3^. Meanwhile, three marker points are also employed and arbitrarily attached on the object surface for more effective and accurate registrations of multi-scanning point cloud dada. Moreover, the workpiece surfaces must be uniformly sprayed with an optical intensifier for enhancing the scanning efficiency and accuracy.

By using above built experimental system, the 3D point cloud data of sampled object can be rapidly collected by multiple scanning operations, then a series of pre-processing operations also need to be conducted in advance, such as filtering noise, repairing holes and calibrating position. Afterwards, through a series of geometrical transformations like translations, rotations and projections, 3D point cloud data can be further converted into more tracTable 2D point cloud data, as plotted in Figure 13a, in which strong randomness and distinct uncertainty can be observed. Finally, the pre-processed 2D point cloud data will be used to rapidly generate the modified DTMs through the triangulation method in Section 2.1, as shown in Figure 13b. Similarly, the statistical analyses are also conducted on experimentally obtained point cloud dataset, more than 46% of data points share six CDTMs, about 93% of data points have five or more CDTMs, as illustrated in Figure 13c. Of course, the established 3D laser scanning system can also output the triangular mesh formats of point cloud data, such as ply, off, stl and obj, which can be directly adopted to developed CDTM-based boundary extraction method and its improved version without any pre-processes and modification described in Section 2.1.

Firstly, original CDTM-based boundary extraction method is conducted on actually collected point cloud dataset with 80, 731 data points, 2179 edge points can be accurately searched with 56 s time consumption, as shown in Figure 14a. To separately investigate the significant influences of distribution characters (such as density and uniformity) and boundary geometrical shapes on the extraction accuracies of different extraction method, the total numbers of miss-judging edge points and error-judging non-edge point must be further analyzed and discussed in detail, especially in the concave and convex regions of workpiece. Therefore, three different local regions are arbitrarily chosen and labeled as area A, area B and area C, respectively, as shown in Figure 14a. In view of more careful observations on different local regions shown in Figure 14b–d, where the distribution of laser-scanned point cloud datasets are characterized by strong randomness and distinct uncertainty, only few miss-judgments of edge points can be observed at whole boundary regions, especially in area C. This is mainly because that the obviously non-uniform and uncertain distribution of point cloud dataset will cause some detrimental deletions and undesirable redundancies of DTMs in certain boundary regions. Nevertheless, a small number of miss-judgments of edge points (FN) can be certainly ignored in most practical applications such as non-contact measurements of mechanical workpieces, this is due to they may have very slight influences on the extraction accuracies in some less important regions. Meanwhile, no error-judgment of non-edge point (FP) can be found throughout whole point cloud dataset of this workpiece, even in the very terrible area C.

As a fair comparison, the popular *α*-shape boundary extractions are also conducted on same point cloud dataset with different *α*-values. From the obtained results shown in Figure 15a, a certain amount of error-judging non-edge points (FP) can be unfortunately found in uneven area C when *α*-value is set as *α* = 0.5, which thus found out more edge points (2271 points) than new CDTM-based methods (2179 edge points). However, these undesirable error-judgments of non-edge points can be completely eliminated in uneven area C when *α*-value increases to *α* = 0.6, 2146 edge points can be extracted with 0.8 s time consumption, but which is slightly less than 2179 edge points of CDTM-based methods, this means that about 33 edge points cannot be found, as shown in Figure 15b. Through the comparative analyses on the results obtained by CDTM-based and *α*-shape methods in local area B, as shown in Figure 14c and Figure 15c, these 33 miss-judging edge points may have a certain level of influence on its extraction accuracies. It should be noted that a higher *α*-value can avoid undesirable error-judgments of non-edge points at a certain extent, but which will conversely cause a certain number of miss-judging edge points. For example, there are about 85 miss-judgments of edge points can be observed when *α*-value increases to 0.7, as shown in Figure 15d. Relative to popular *α*-shape method, all obtained results distinctly indicate that CDTM-based boundary extraction method can more precisely and robustly describe the boundary feathers of unorganized point cloud data with strong randomness and distinct non-uniformity, especially for complex-shaped objects with holes and cavities.

Finally, the statistically improved CDTM-based boundary extraction method will be also conducted on same point cloud data when CDTM numbers are selected as *δ_N_* ≤ 3, *δ_N_* ≤ 4, *δ_N_* ≤ 5 and *δ_N_* ≤ 6, respectively, the total numbers of filtered data points (*N*_0_′) and edge points (*N*_1_) can be directly obtained, their corresponding edge extraction accuracies *ε* and time consumptions *T_c_* are further calculated according to Equation (10), as well as fairly comparing with original CDTM-based method (*δ_N_* = All) and α-shape (*α* = 0.6) method, then all obtained results are illustrated in Figure 16 and listed in Table 3.

When CDTM number is set as *δ_N_* ≤ 3, the improved CDTM-based method can find out 1425 edge points with 2.5 s time consumption, but its corresponding edge extraction accuracy *ε* is less than 65%, which cause a massive loss of boundary geometric features, as shown in Figure 16a. When CDTM number is selected as *δ_N_* ≤ 4, 2037 edge points can be rapidly extracted with 5.2 s time consumption, corresponding edge extraction accuracy *ε* sharply grows to 93.5%, as shown in Figure 16b. Similarly, 2167 edge points can be found when CDTM number is selected as *δ_N_* ≤ 5, at this moment, the edge extraction accuracy *ε* perfectly increases to 99.4%, corresponding time consumption (16.0 s) still is acceptable relative to original CDTM-based method. Eventually, all edge points (2179 points) can be entirely extracted when CDTM number is selected as *δ_N_* ≤ 6, namely its edge extraction accuracy *ε* is perfectly equal to 100%, but corresponding time consumption reaches 41.0 s that is 73.2% of CDTM-based method (*δ_N_* = All). Known from above comparing analyses, the improved CDTM-based method can optimally select CDTM number to be *δ_N_* ≤ 4 or *δ_N_* ≤ 5 in consideration of the practical demands on efficiencies and accuracies in different application scenarios. In summary, all experiment results can well demonstrate that both the CDTM-based boundary extraction method and its improved version can efficiently and precisely describe the boundary features of unorganized point cloud data that have to suffer strong randomness and distinct uncertainty, but original CDTM-based method needs length calculating time than its improved version when they have an approximate extraction accuracy, so that the statistically improved CDTM-based method will be more highly suggested in various industrial applications like on-line dimension measurements of mechanical parts, which generally demand an excellent real-time performance.

## 5. Conclusions

For efficiently and robustly characterizing the boundary geometries of unorganized point cloud data, this study proposed a novel type of boundary extraction method based on concurrent Delaunay triangular meshes (CDTMs), called as CDTM-based boundary extraction method. Firstly, traditional Delaunay triangular meshes of unorganized point cloud data were modified with several simple but effective regulations for removing the undesirable triangular meshes with abnormal sizes and irregular shapes, which mainly appear in the concave substructures like holes and cavities. Secondly, the basic principles of new CDTM-based method were mathematically constructed based on Euler distances and cosine theorem, then its statistically improved version was also developed for higher extraction efficiency, namely statistical CDTM-based boundary extraction method, which will strictly filter out most potential non-edge points in advance based on the statistical analyses on CDTM numbers of each data point. Finally, these two CDTM-based methods and popular *α*-shape method were adopted to conduct a series of boundary extractions on several point cloud datasets of complex-shaped workpieces to investigate their main performances, such as the time consumption and extraction accuracy. Several dominant conclusions about this study can be briefly summarized as below:

(a) All obtained results can distinctly indicate that these two proposed CDTM-based boundary extraction methods can precisely extract the boundary features of unorganized point cloud data with strong randomness and distinct uncertainty. Meanwhile, very few miss-judgments of edge points and no error-judgment of non-edge point can be observed. Note that the statistical CDTM-based method with a suitable CDTM number *δ_N_* will cost much less time consumption than its original version when their extraction accuracies are very approximate, which will therefore be highly recommended in various practical applications, such as on-line size measurements of mechanical parts.

(b) The statistical analyses were conducted on four different point cloud datasets of complex-shaped workpieces, respectively, then all obtained results distinctly show that almost all of edge points have four CDTMs or less, but most of non-edge points share five CDTMs or more. For all sampled workpieces or objects, near 50% or even more data points have six CDTMs, more than 98% of edge points share less than CDTMs. These valuable statistical analyses are the important foundations to strictly filter out majority of potential non-edge points, further greatly decreasing time consumption.

(c) Comparing with *α*-shape method, these two CDTM-based methods both possess higher extraction accuracies. This is mainly because that too high or too low α-values may adversely cause few miss-judgments of edge points or error-judgments of non-edge points. In view of *α*-shape method must troublesomely optimize its α-value according to random and uneven distributions of unorganized point cloud data, but the CDTM-based method and its improved version have better operability and stronger robustness since they have no extra threshold that needs to be hardly selected.

(d) The *α*-shape method can directly call the built-in functions in MATLAB 2015b, thus its time consumption is less than two CDTM-based methods. However, they should have the near or even less time consumption due to similar time complexities. It should be noted that these two CDTM-based methods can be only applied to extract the planar boundary features of 2D point cloud datasets in this study, but which already can meet the requirements in most application scenarios after conducting 3D-to-2D projection and transformation on spatial 3D point cloud data.

## Figures and Tables

**Figure 1 sensors-23-01915-f001:**
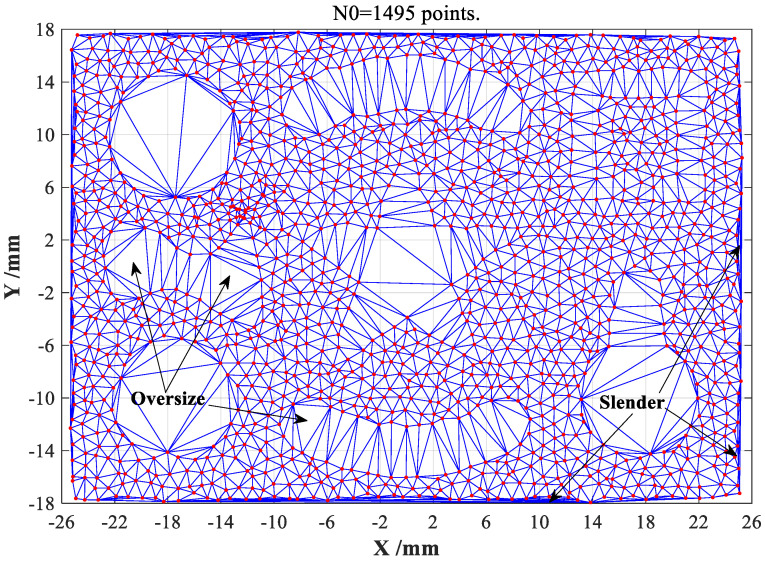
The traditional Delaunay triangular meshes (DTMs) for a complex-shaped object.

**Figure 2 sensors-23-01915-f002:**
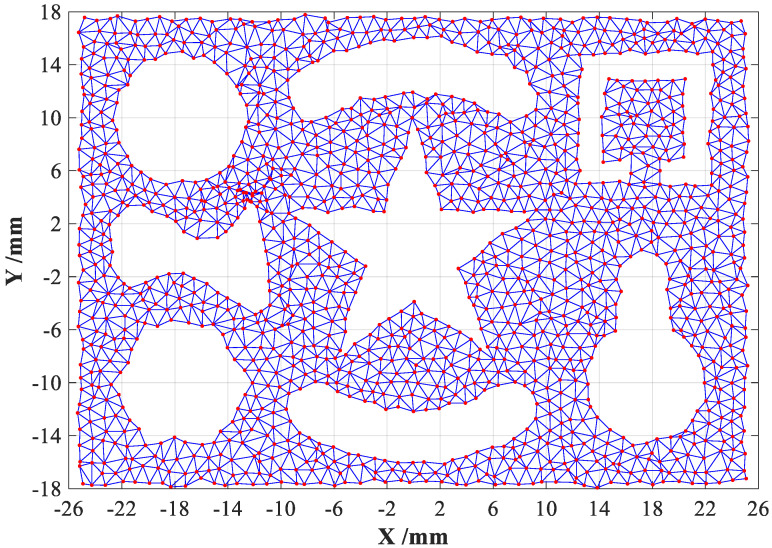
The modified Delaunay triangular meshes (DTMs) for a complex-shape object.

**Figure 3 sensors-23-01915-f003:**
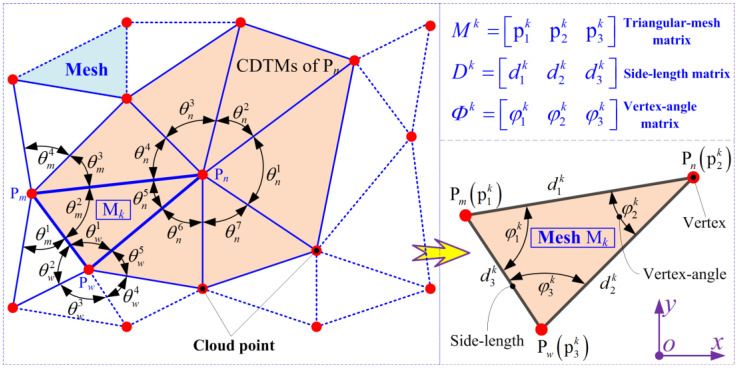
Basic principles of the CDTM-based boundary extraction method for point clouds.

**Figure 4 sensors-23-01915-f004:**
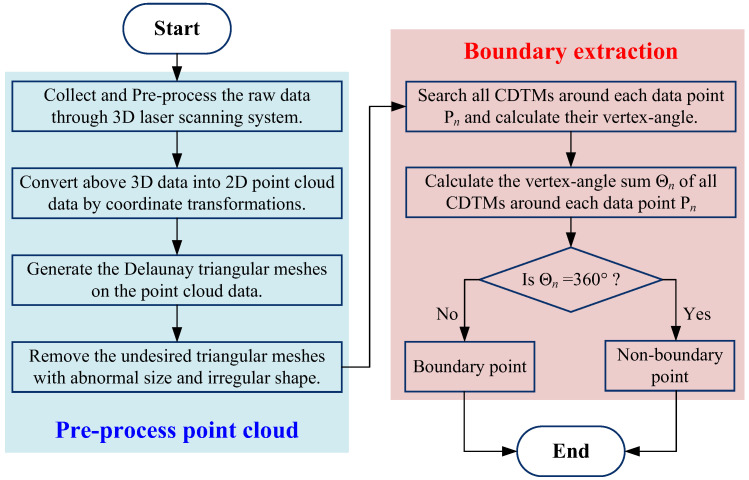
The flowchart of the developed CDTM-based boundary extraction method.

**Figure 5 sensors-23-01915-f005:**
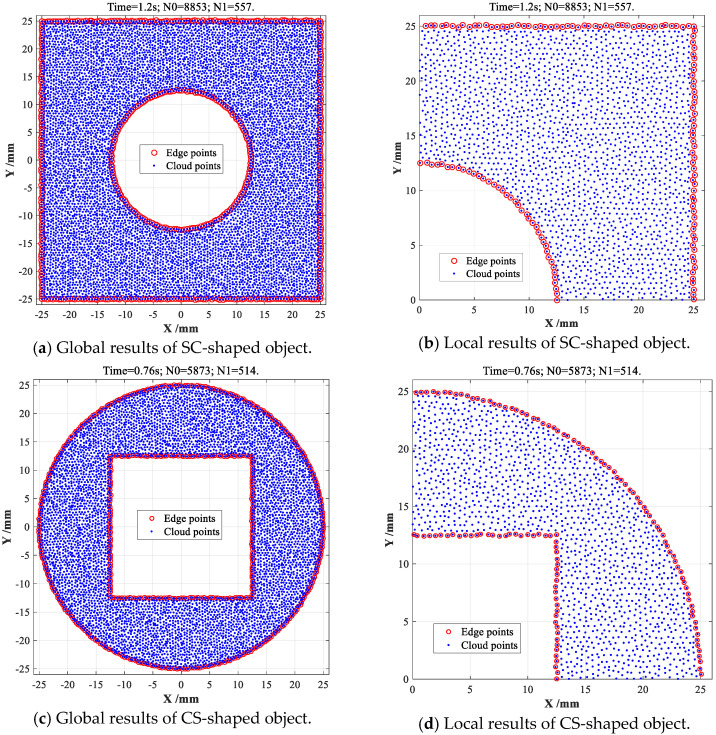
Boundary extractions on two different point cloud datasets with CDTM-based method.

**Figure 6 sensors-23-01915-f006:**
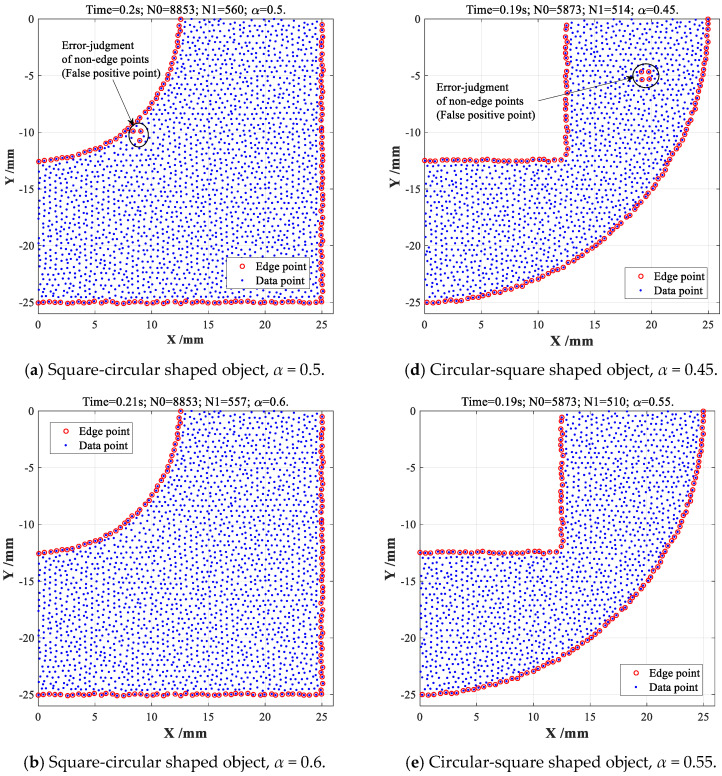
Boundary extractions on two different point cloud datasets with *α*-shape method.

**Figure 7 sensors-23-01915-f007:**
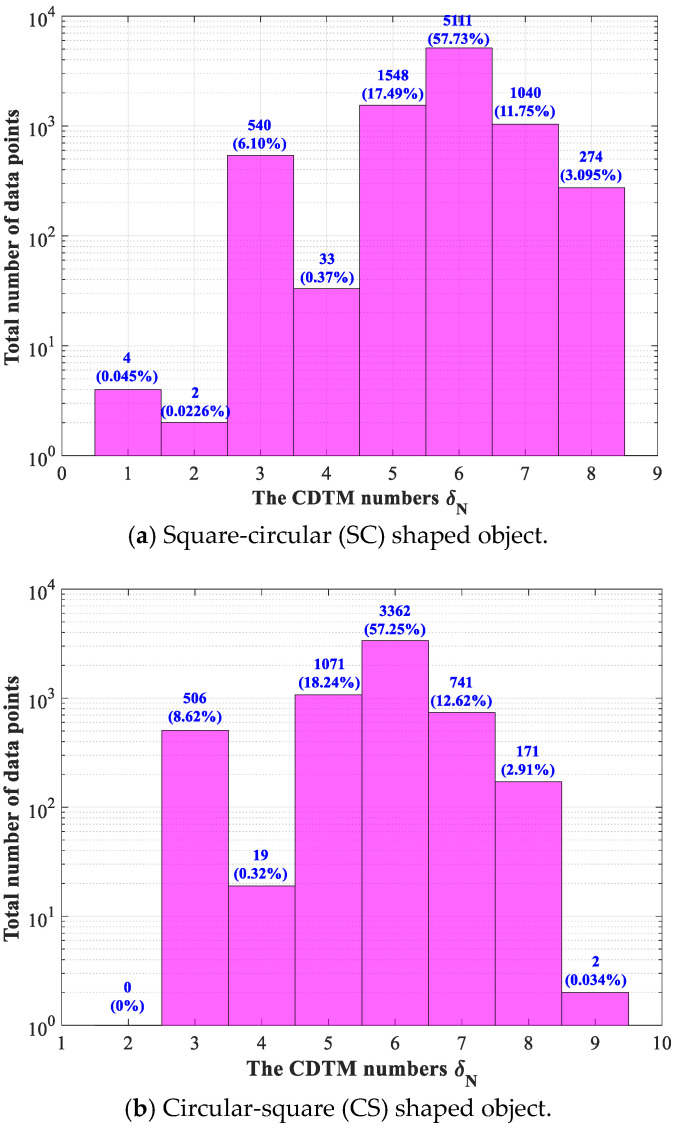
Statistical analyses on the total numbers of data points with different number of CDTMs.

**Figure 8 sensors-23-01915-f008:**
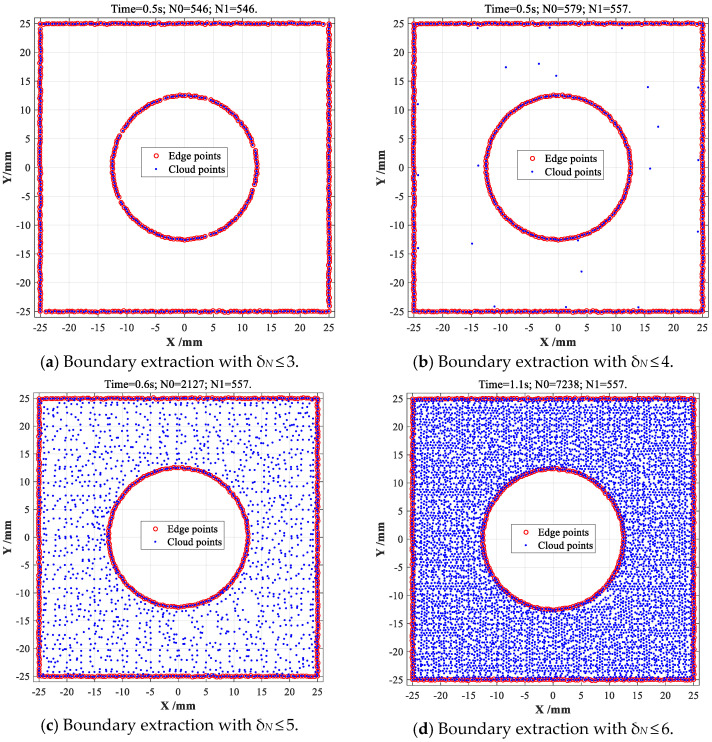
Statistical CDTM-based boundary extraction on SC-shaped point cloud with different δ*_N_*.

**Figure 9 sensors-23-01915-f009:**
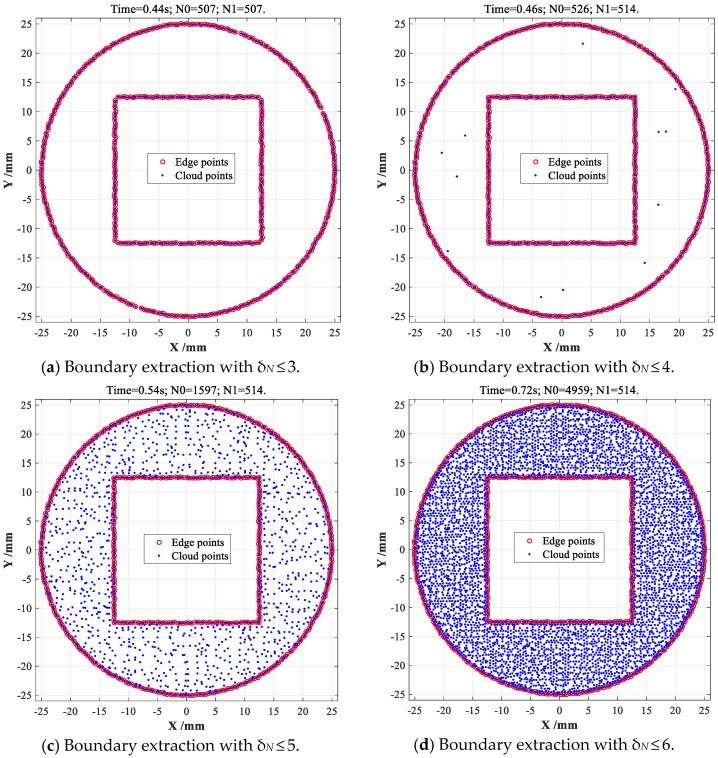
Statistical CDTM-based boundary extraction on CS-shaped point cloud with different *δ_N_*.

**Figure 10 sensors-23-01915-f010:**
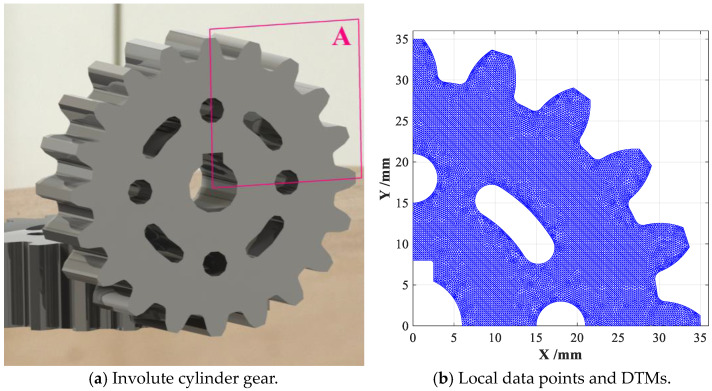
The adopted involute cylinder gear and its generated DTMs.

**Figure 11 sensors-23-01915-f011:**
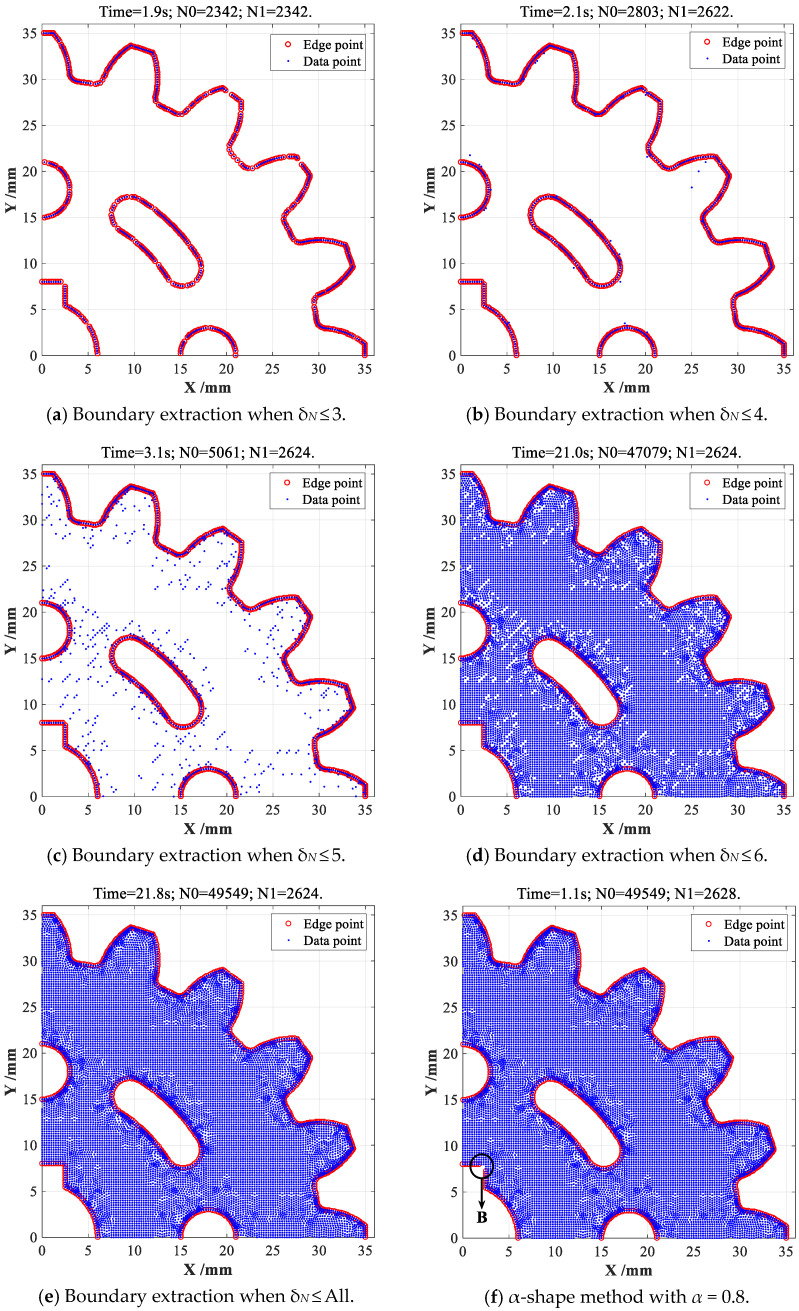
Boundary extractions and performance comparison on gear object under different *δ_N_*.

**Figure 12 sensors-23-01915-f012:**
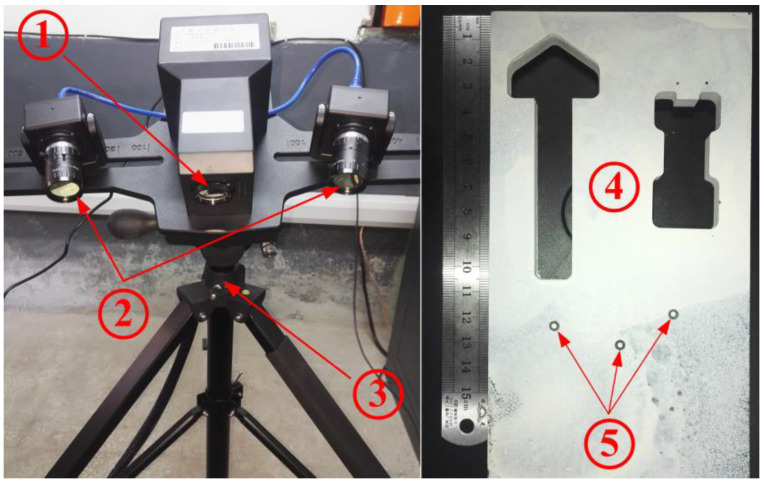
The photograph of experimental setup for collecting point cloud data of complex-shaped object. 1—Structured light laser; 2—Double cameras; 3—Tripod platform; 4—Complex workpiece; 5—Marked points.

**Figure 13 sensors-23-01915-f013:**
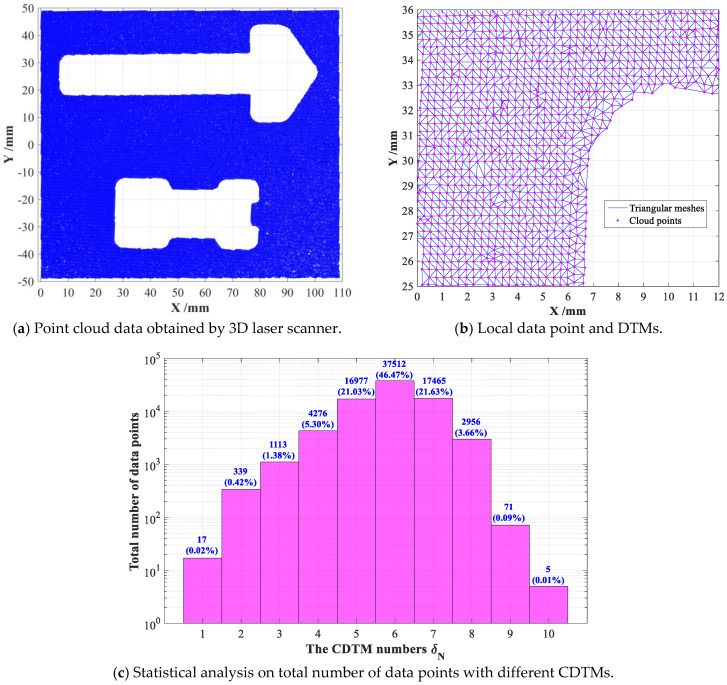
The pre-processed point cloud data and their modified Delaunay triangular meshes.

**Figure 14 sensors-23-01915-f014:**
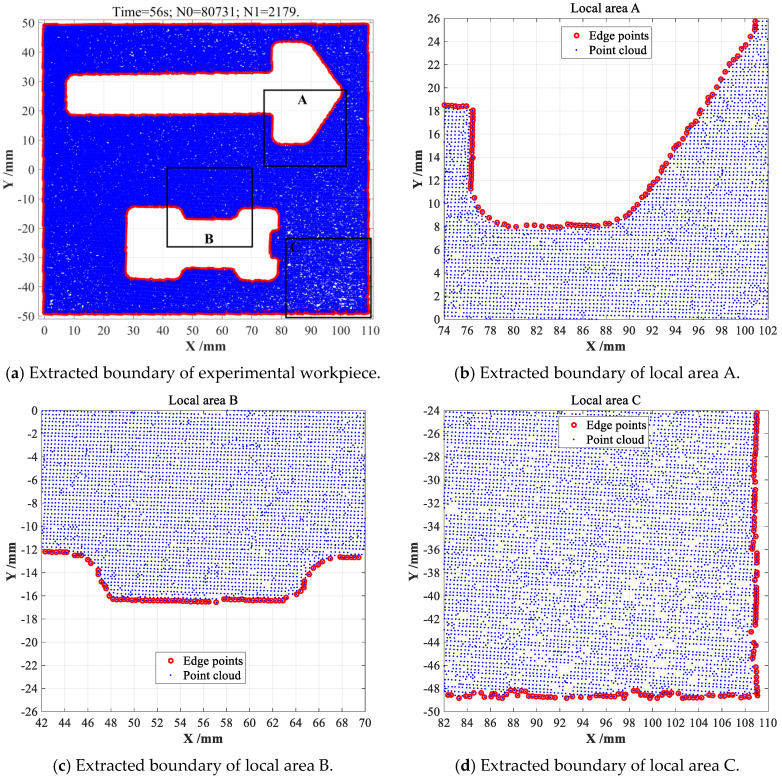
The CDTM-based boundary extraction experiments on complex-shaped point clouds.

**Figure 15 sensors-23-01915-f015:**
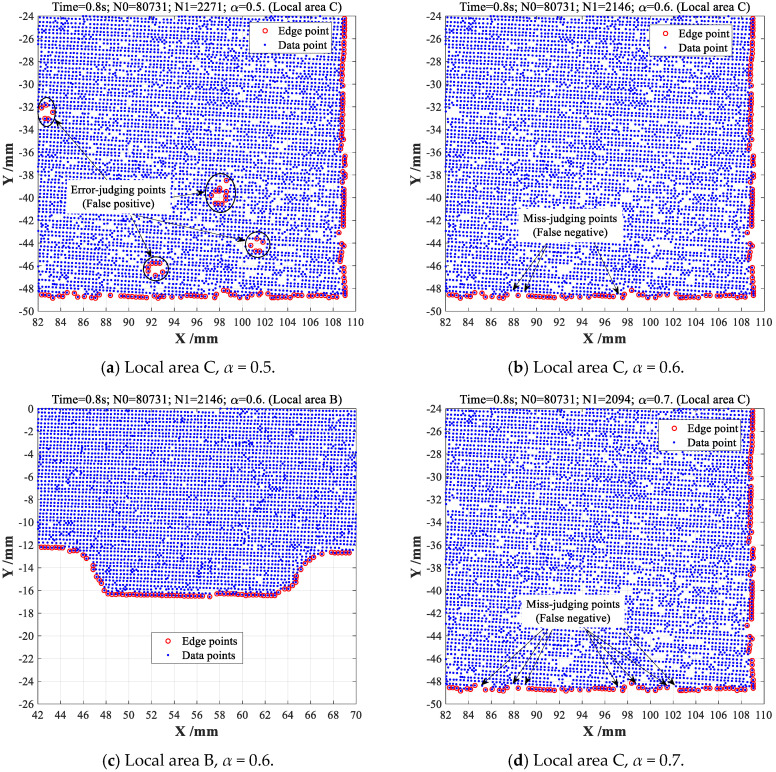
The α-shape boundary extraction experiments on complex-shaped point clouds.

**Figure 16 sensors-23-01915-f016:**
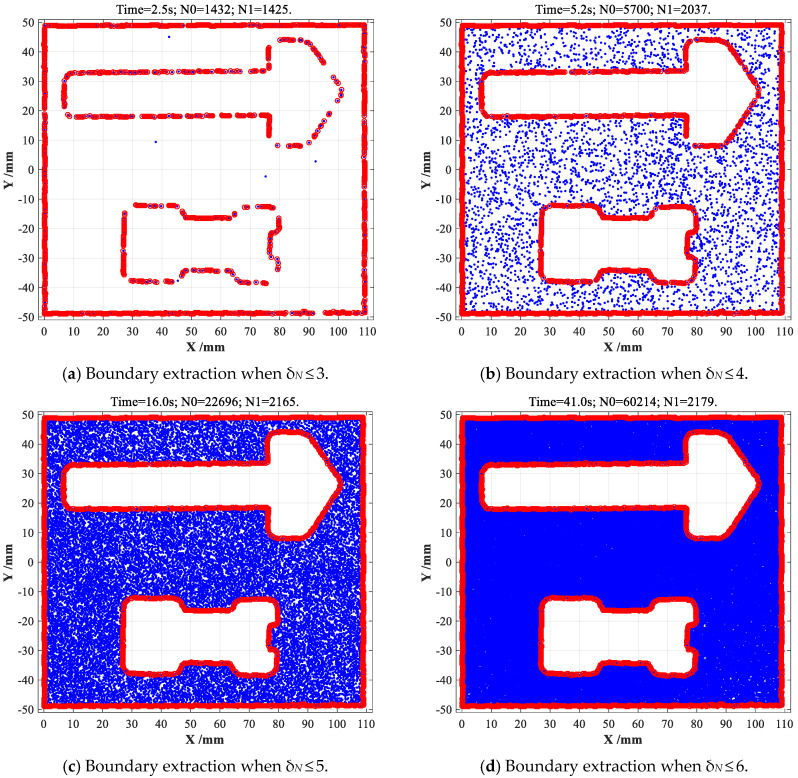
The statistical CDTM-based boundary extraction on experimentally laser-scanning point cloud data of a complex-shaped mechanical parts.

**Table 1 sensors-23-01915-t001:** Comparison analyses on the CDTM-based and α-shape boundary extraction results of SC-shaped and CS-shaped point clouds with different *δ_N_*. (*α*_sc_ = 0.6 and *α*_cs_ = 0.55).

Objects	Performance Index	*δ_N_* ≤ 3	*δ_N_* ≤ 4	*δ_N_* ≤ 5	*δ_N_* ≤ 6	*δ_N_* = All	*α*-Shape
SC-shaped	Time consumption *T_c_*	0.5 s	0.5 s	0.6 s	1.1 s	1.2 s	0.21 s
	Number of data point *N*_0_′	546	579	2127	7238	8853	8853
	Number of edge point *N*_1_	546	557	557	557	557	557
	Edge extraction accuracy *ε*	98.0%	100%	100%	100%	-----	100%
CS-shaped	Time consumption *T_c_*	0.44 s	0.46 s	0.54 s	0.72 s	0.76 s	0.19 s
	Number of data point *N*_0_′	507	526	1597	4959	5873	5873
	Number of edge point *N*_1_	507	514	514	514	514	510
	Edge extraction accuracy *ε*	98.6%	100%	100%	100%	-----	99.2%

**Table 2 sensors-23-01915-t002:** The performance comparison among different boundary extraction method conducted on the point cloud data of involute cylinder gear.

Performance Index	*δ_N_* ≤ 3	*δ_N_* ≤ 4	*δ_N_* ≤ 5	*δ_N_* ≤ 6	*δ_N_* = All	*α* = 0.8
Time consumption *T_c_*/s	1.9 s	2.1 s	3.1 s	21.0 s	21.8 s	1.0 s
Total number of dada point *N*_0_′	2342	2803	5061	47,079	49,549	49,549
Total number of edge point *N*_1_	2342	2622	2624	2624	2624	2628
Edge extraction accuracy *ε*	89.25%	99.92%	100.0%	100.0%	-------	100.2%

**Table 3 sensors-23-01915-t003:** The performance comparison among different boundary extraction method conducted on the actual point cloud data of experimental workpiece.

Performance Index	*δ_N_* ≤ 3	*δ_N_* ≤ 4	*δ_N_* ≤ 5	*δ_N_* ≤ 6	*δ_N_* = All	*α* = 0.6
Time consumption *T_c_* /second	2.5 s	5.2 s	16.0 s	41.0 s	56.0 s	0.8 s
Total number of dada point *N*_0_′	1432	5700	22,696	60,214	80,731	80,731
Total number of edge point *N*_1_	1425	2037	2165	2179	2179	2146
Edge extraction accuracy *ε*	64.8%	93.5%	99.4%	100%	------	98.5%

## Data Availability

Not applicable.

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
