# Peer review of "A Novel Type of Boundary Extraction Method and Its Statistical Improvement for Unorganized Point Clouds Based on Concurrent Delaunay Triangular Meshes"

_sensors, 2023, doi:10.3390/s23041915_

Round 1
Reviewer 1 Report
Dear Authors.
Unfortunately, your paper (A novel type of boundary extraction method and its statistical improvement for unorganized point clouds based on concurrent Delaunay triangular meshes) is not recommended for publication. The main reasons given:
#1. Code and Simulation
What kind of Simulation/Coding language (version) did you use?
-Please add/more version and description.
-Simulation/Coding language environment.
-Environment explanation/presentation more.
#2. This paper is general and has a low level of Experiment results and analysis.
#3. Introduction
The "Introduction" information presented is not new.
I recommend additional/rewrite the "Introduction".
-Need a nice story of Introduction.
#4. Methods.
-They should be described with sufficient detail to allow others to replicate and build on published results. New methods and protocols should be described in detail while well-established methods can be briefly described and appropriately cited. Give the name and version of any software used and make clear whether computer code used is available. Include any pre-registration codes (Methods).
#5. Results
Results need clearly.
-Need nice story of results.
#6. Other
Scientific Soundness: Low.
#7. English
-Moderate English changes required.
#8. Conclusion
-You need to write more of the conclusion part.
-Future work" write more. (Must be improved)
#9.
Does it have something to do with the mdpi Sensor journal?
But, the strength of the paper included:
-the topic is interesting.
Author Response
See the appendix for details.

Reviewer 2 Report
The proposed work addresses a core problem in various types of segmentation tasks, or similarly where the boundary unorganised point clouds lie, for configurations that may include ‘holes’. They recommend a criterion to guide the extraction of these boundaries, which is based on the interior angles around the centre of a common-vertex. A statistical utilisation of this criterion leads to a proposal of a boundary extraction method. The aimed theoretical contribution is significant as these results may have application not only in 3D, but in 2D (as illustrated in several figures), but also in higher dimensional spaces.
Albeit the paper is submitted to the industrial section, the scope of the application of this work is not clear from the beginning. A minor problem in the introduction is that it talks about laser scanning, which is connotated with Time-of-Flight scanning. It is not clear if the VTOP 200B, 392 is a multi-ocular stereo system with a laser projector or a Time-of-Flight sensor.
The theoretical section uses one illustrative example, which is fine. In the experiments with real data, one image is tested. Several synthetic datasets can be generated with simple means, to support statistical analysis of the proposed contribution. If the focus is on the industrial part, then the experimental validation should be expanded so that it can be quantitatively studied in a more substantial population of tests. Also, several open data sets could serve this purpose, e.g. Middlebury Stereo etc.
Last but not least the experimental evaluation does not provide any comparative analysis with other works, even though several of them are mentioned in the literature review. It is understood that the code of these methods is not available that you cannot implement them. However, running your method, comparing, and discussing your findings on the same data (or at least similar examples) is necessary for reporting the advantages of the proposed work over others that can be found in the literature.
Author Response
See the appendix for details.

Round 2
Reviewer 1 Report
Dear Authors.
Unfortunately, your paper is not recommended for publication. The main reasons given:
This paper was not improved considering reviews.
Author Response
See attachment for details.

Reviewer 2 Report
I find that the authors properly responded to the comments of the reviewers. There is a bit of a problem with the submitted manuscript, in that it has "broken links". That is references inside the paper read "Error! Reference source not found" which I suppose is an MS Word error that can be fixed in editing. This is the only reason I suggest minor revisions.
Author Response
See attachment for details.
